# Effects of the COVID-19 Restrictions on Eating Behaviour and Eating Disorder Symptomology in Female Adolescents

**DOI:** 10.3390/ijerph19148480

**Published:** 2022-07-11

**Authors:** Lois Muth, Karl-Heinz Leven, Gunther Moll, Oliver Kratz, Stefanie Horndasch

**Affiliations:** 1Department of Child and Adolescent Mental Health, Friedrich-Alexander-Universität Erlangen-Nürnberg, Schwabachanlage 6, 91054 Erlangen, Germany; lois.muth@fau.de (L.M.); gunther.moll@uk-erlangen.de (G.M.); oliver.kratz@uk-erlangen.de (O.K.); 2Institute for the History and Ethics of Medicine, Friedrich-Alexander-Universität Erlangen-Nürnberg, Glückstr. 10, 91054 Erlangen, Germany; karl-heinz.leven@fau.de

**Keywords:** anorexia nervosa, depression, adolescents, COVID-19 pandemic

## Abstract

Confinement due to the COVID-19 pandemic imposes a burden on adolescents worldwide and may seriously impact patients with an eating disorder (ED). The current *FR*anconian *A*norexia *N*ervosa during *CO*VID-19 (FRANCO) study explored (1) perceived change of depressive and ED symptomology during lockdown, (2) the role of social media, and (3) coping strategies of anorexia nervosa (AN) patients and clinical as well as healthy comparison groups. From June 2021 to September 2021, 222 female adolescents (19 with AN, 20 with depression, 45 with a self-reported psychiatric disorder (SRPD), and 138 controls) aged 11.2 to 18.9 years completed a one-time anonymous survey retrospectively reporting back on ED and depressive symptomology before and during the pandemic, the impact of social media, and coping strategies. A reduced quality of life (QoL) due to confinement was observed in almost half of female adolescents. All groups reported a significant perceived increase of disordered eating, overeating, anxiety, and depressive symptoms and emotion-regulation problems. In AN patients, significantly higher percentual deterioration of disordered eating and anxiety and depressive symptoms was found. For controls, a younger age and higher susceptibility of the sociocultural body image significantly correlated with increased disordered eating. Large-scale media literacy interventions are recommended.

## 1. Introduction

Although numerous restrictions have been put in place to limit the spread of COVID-19, the pandemic remains dominant to this day. Social distancing, school closures, and cancellation of leisure activities may impact the mental health of adolescents, and those with an eating disorder (ED) may face unique risks. EDs are particularly common in adolescents, especially in girls [1]. The representative Germany-wide COPSY (impact of *CO*VID-19 on *psy*chological health in children and adolescents) study found that restrictions due to the COVID-19 pandemic have led to an increased risk for mental health problems in children and adolescents, with 40% of adolescents reporting a reduced quality of life (QoL) in the first wave of the pandemic [2] and 48% in the second wave of the virus [3] as opposed to 33% pre-pandemic [4]. A longitudinal UK study reported an increase of depression symptoms in children and adolescents compared to the time period before lockdown [5]. Shortly after this, a longitudinal study found that conduct problems and emotional symptoms increased particularly in children and less in adolescents during early lockdown [6]. While Ravens-Sieberer et al. [3] found that female adolescents had fewer problems with mental health, emotion, conduct, hyperactivity, and anxiety during the second wave of the pandemic than males, a systematic review of 23 studies and a longitudinal Icelandic study found that mental health deteriorated particularly in female adolescents compared to males [7,8]. Interestingly, a longitudinal study among German adolescents found no increase in mental health-related issues but a decrease in suicide plans and conduct problems, stating that the COVID-19 restrictions had no effects or even had positive effects on adolescents’ mental health [9]. However, these results do not rule out that there are particularly vulnerable groups who might be affected strongly.

During the COVID-19 pandemic, an increase in ED symptomology was observed in people with anorexia nervosa (AN) [10]. Analysing admission and readmission rates, a study found a significant increase in inpatient count for adolescents with AN after the COVID-19 restrictions, with one-third reporting primary correlations between the pandemic and their AN [11]. To name a few, higher admission rates were found for adolescents with ED in North America [11,12,13], Australia [14], and Israel [15] and for adults with ED in the UK [16] and in New Zealand [17]. Overall, 40% of inpatient adolescents with ED described the pandemic as a worsening factor for their ED [13]. In the same vein, 63% of adolescents with ED reported a “triggering environment” due to the pandemic as cause for worsening of their ED symptomology [18]. Adolescents who reported the worsening of their ED due to the COVID-19 pandemic showed an 18-times decreased motivation to recover, a 24-times increase of ED thoughts [18], and higher rates of medical instability (79% vs. 55%) [13]. Adolescents with AN who were hospitalized during COVID-19 lockdown showed 8-times higher rates of readmission within 30 days after discharge than those hospitalized pre-pandemic [11]. In contrast to this, a Spanish study found that AN patients had a relatively stable body mass index (BMI), even gaining weight, and that disordered eating and emotion dysregulation significantly decreased for people with AN during confinement [19].

Especially younger people show an addictive use of technology, with the female gender significantly linked to addictive use of social media [20]. The use of social networking sites, especially when appearance-based, is associated with body dissatisfaction [21,22,23] and a low self-esteem [24,25]. Social media promotes exercising and healthy food, which might function as a trigger for disordered eating. More time per day spent on social media is significantly associated with increased eating concerns in young adults [26]. A study found a significant increase in use of social network sites during lockdown, with a significant relationship between the rate of Instagram use and an increase in body dissatisfaction, drive for thinness, and a decreased self-esteem in young women [27]. During the pandemic, exposure to triggering social media messages that promote losing weight and getting fit were mostly described as a factor for worsening of ED symptomology by people with AN, thus representing a more important factor than for people with other ED [28].

Fernández-Aranda et al. [29] suggested coping strategies such as creative activities, communicating, and light exercise [29]. Not long after this, a study by Schlegl et al. [10] studied the effectiveness of these coping strategies and found that mindfulness and the opportunity to learn things they never found the time to before were the most helpful among adolescents [10].

Few previous studies have explored how AN patients report back on times of confinement during the pandemic compared to other groups of adolescents. The aims of the present study were to explore: (1) different areas of life, including depressive symptoms and ED symptomology; (2) the role of social media; and (3) coping strategies of female adolescents before vs. during lockdown according to self-reported retrospective inquiry. To differentiate between subgroups of adolescents and check for specificity, we included two clinical samples (adolescents suffering from AN and adolescents suffering from a depressive disorder) as well as adolescents from a community sample.

**H1.** 
*We hypothesize that clinical groups (AN, depression) report a higher increase in anxiety and depressive symptoms and emotion regulation problems during lockdown than the other groups and that individuals with AN describe a higher increase of weight concerns and restrictive eating during lockdown than the other groups.*


**H2.** 
*We hypothesize that individuals with AN are more likely to show an elevated perceived pressure, awareness, and internalization of the sociocultural body ideal and that the extent will correlate with the exacerbation of disordered eating during the pandemic.*


**H3.** 
*We hypothesize that a higher number and helpfulness of coping strategies used correlates with a better outcome of ED and emotional symptoms during the pandemic.*


## 2. Materials and Methods

### 2.1. Participants

The data collection of this *FR*anconian *A*norexia *N*ervosa during *CO*VID-19 (FRANCO) study was conducted in Germany between June and September 2021. Schools were either open or on summer break. When reporting on times of “lockdown”, we asked participants to think back to a time when schools were closed due to confinement measures. During the “first lockdown” in March and April 2020, schools closed for the first time. This was followed by limited capacities and the introduction of hygiene rules at schools. Schools closing due to confinement measures had most recently been the case from early January 2021 to mid-March 2021 due to a nationwide “second lockdown”, which included school closures, curfew, and rules on the limitation of numbers of social contacts. Following this, schools were open or closed depending on the local incidence rates, which lowered by the end of May 2021.

Our sample comprised 222 female participants recruited at the Department of Child and Adolescent Mental Health at the University Clinic Erlangen and online:Clinical sample19 patients with AN;20 patients with depression.Online sample45 adolescents with a self-reported psychiatric disorder (SRPD);138 typically developing control participants.

The AN and depression samples were diagnosed by an experienced child and adolescent psychiatrist or psychologist and received their diagnosis according to ICD-10 (AN or atypical AN (ICD-10: F50.0 or F50.1), major depressive disorder, or adjustment disorder with depressed mood (ICD-10: F32.X or F43.21), World Health Organization, 1993) and were asked by their therapist if they were willing to voluntarily complete the anonymous study questionnaire. The online groups (SRPD, controls) were recruited online through social media, email, and social contacts to complete the anonymous survey online. This online sample was later divided into groups based on self-reported disclosure of a pre-existing psychiatric diagnosis, which includes EDs (e.g., AN, BN) and others (e.g., anxiety, depression, phobia, obsessive-compulsive disorder). Thus, the SRPD group was made up of adolescents who reported a pre-existing psychiatric diagnosis, and the controls comprise adolescents who stated they had no pre-existing psychiatric diagnosis. Informed consent was obtained from all subjects involved in the study.

Excluding criteria were acute psychotic symptomology, impairment of intelligence, or insufficient knowledge of the German language.

### 2.2. Survey Procedures and Questions

The anonymous questionnaire was completed by all participants.

The main part of the survey was based on the COVID Isolation Eating Scale (CIES), a newly created instrument for measuring the effects of confinement for patients with EDs. CIES obtained adequate goodness-of-fit in adolescents and adults (mean age 34 years) comparing different EDs (e.g., AN, BN) and obesity to one another [19]. Four CIES factors were incorporated. F1 (10 items) measured disordered eating symptomology such as weight concerns and strategies to reduce food intake (restrictive eating, bingeing/purging, and use of laxatives and diuretics). F2 (10 items) focused on eating-related style, mainly overeating. F3 (11 items) assessed anxiety and depressive symptoms including issues with sleep, negative thoughts, social isolation, health concerns, and sexual problems. F4 (5 items) covered problematic emotion regulation including emotional control, aggression and feelings of anger, shame, and irritability. The scale consists of two 5-point Likert scales ranging from “never/almost never/sometimes” to “almost always/always”. One CIES scale is directed at “Before the restrictions” and one at a time with more restrictions as well as “Now”, which in our case, due to the retrospective nature of our survey, were titled “Before ‘lockdown’” and “During ‘lockdown’”.

All participants completed the Sociocultural Attitudes Towards Appearance Questionnaire (SATAQ), which assesses the impact of sociocultural factors on body image. SATAQ was developed to assess recognition and acceptance on societal standards of appearance and has been widely employed in women and adolescent girls [30]. The German version, called SATAQ-G, that was employed is a reliable and valid tool to analyse influences of sociocultural body ideals on body image [31]. It consists of 16 items and three subscales: pressure, awareness, and internalization. The response options on a 5-point Likert scale range from “strongly disagree” to “strongly agree”. Instead of the influence of TV and magazines, we were interested in the impact of social media; thus, items 1–5, 6 and 7 were rephrased, replacing “magazines” and “TV” with “social media”.

The questionnaire contained 21 coping strategies from a survey on the impact of the COVID-19 pandemic on adolescent and adult patients with EDs [10], which are based on a suggestion by Fernandez-Aranda et al. [19]. The response options range from “not used” (0) to “not helpful” (1) and “very helpful” (5). A statement on the perceived QoL since the implementation of restriction measures (lockdown) was included: “Lockdown has deteriorated my quality of life”, with a 5-point Likert scale ranging from “strongly disagree” to “strongly agree” [10]. Finally, participants were given the opportunity to remark on “personal experiences and helpful strategies during lockdown”.

The present study was approved by the Ethics Committee of the University Hospital of Erlangen and was conducted in accordance with the Declaration of Helsinki, and the protocol was approved by the Institutional Ethics Committee of University Hospital Erlangen (protocol code 250_21 B). The data presented in this study are available on request from the corresponding author.

### 2.3. Data Analysis and Statistics

Statistical analyses were carried out with IBM SPSS Statistics^®^ (version 27, IBM, Armonk, NY, USA).

A one-way analysis of variance (ANOVA) was run with age and BMI as dependent variables and the groups as factors. Post hoc *t*-tests for positive effects of “group” with Bonferroni corrections for the number of tests were applied.

Since the CIES factors vary in number of items, percentual differences between self-reported measures of the CIES factors before and during lockdown were calculated for better comparability. To examine the percentual changes of the CIES factors, we conducted one-way ANOVAs (participant group as the between-group factor) with post hoc Bonferroni-corrected *t*-tests for number of tests.

A one-way ANOVA was run with the scores of SATAQ categories “pressure”, “awareness”, and “internalization” as dependent variable and the groups as factor. A post hoc multiple comparisons *t*-test with Bonferroni corrections for multiple testing was carried out to pairwise compare groups. In order to evaluate the relationship between percentual changes of disordered eating and the degree of susceptibility to the sociocultural body image through media, correlations between the SATAQ scores and changes of F1 were run with Bonferroni corrections for number of tests.

To measure the percentual use and helpfulness of coping strategies between groups, a one-way ANOVA with post hoc Bonferroni corrections for number of tests was applied. Correlations with Bonferroni-corrected significance levels were carried out between the number and helpfulness of the coping strategies and percentual changes in each CIES factor.

Qualitative data were gathered when participants were given the opportunity to write down remarks on “personal experiences and helpful strategies during lockdown” at the end of the survey. A thematic analysis was conducted as described by Braun and Clarke [32]. The thematic analysis was used to identify meaningful patterns in the participant’s written responses in order to provide direct quotes grouped into themes. The aim was to present data-driven, descriptive results. First, all responses were read repeatedly to gain an overall understanding, making sure to keep an open mind towards the experiences of the participants. A straightforward meaning between language and experience was assumed, applying a realist and semantic approach. Each response was divided into thematic sections, and inductive, data-driven categorization by meaning was employed by the first (L.M.) and last author (S.H.), resulting in two main categories (“personal experiences” and “strategies”), each with their own themes. Themes were defined as collections of meaning units connected by a similar content. Initial themes were reviewed until each of them described a unique point of data. They were reviewed collaboratively by the two reviewers to ensure consensus. Exemplary quotes that illustrate the content of each theme were chosen and translated from German into English, staying as close to the original meaning as possible.

## 3. Results

### 3.1. Participant Characteristics

Of the 45 participants with a SRPD, 8 reported at least one ED (e.g., AN, bulimia nervosa), 30 at least one other psychiatric disorder (e.g., anxiety, depression), and 7 both at least one ED and at least one other psychiatric disorder.

All participants were female and between 11.2 and 18.9 years old. There was a significant difference in age between the groups; F(3,218) = 4.164, *p* < 0.01. Thus, adolescents with SRPD were significantly older than the AN group (*p* < 0.01). The groups differed significantly in BMI; F(3,218) = 7.514, *p* < 0.001. The AN group presented with a significantly lower BMI than all other groups (depression, SRPD: *p* < 0.001, controls: *p* < 0.01). Table 1 contains sociodemographic information.

### 3.2. Quality of Life

Of all participants, almost half (48%) reported a deterioration of their QoL due to the COVID-19 restrictions. QoL deteriorated in 70% of adolescents with depression, in 68% of adolescents with AN, in 53% of adolescents with a SRPD, and in 40% of the controls (Table 2).

### 3.3. Changes of CIES Factors

In all groups, retrospectively self-reported measures of all CIES factors (F1–F4) before vs. during lockdown significantly increased (Appendix A). Large effect sizes (|d| > 0.8) were observed in the absolute increase of F1 in AN patients (|d| = 2.60); F3 in the depression (|d| = 1.00) and control group (|d| = 1.17); and F4 in the AN (|d| = 1.45), depression (|d| = 1.13), and control group (|d| = 0.91).

The CIES factors vary in number of items. Thus, for better comparison, percentual changes were calculated (Figure 1 and Appendix A). Between-group comparison of percentual changes showed significant differences between the groups in F1 (F(3,218) = 18.406, *p* < 0.001); F3 (F(3,218) = 7.770, *p* < 0.001); and F4 ((3,218) = 3.463, *p* < 0.05). View Figure 1 for result of Bonferroni-corrected post hoc *t*-tests. A significant Bonferroni-corrected correlation between age and F1 in the control group (*p* < 0.01) indicates that younger controls reported a higher percentual increase of disordered eating than older controls.

### 3.4. Social Media Use

SATAQ was applied to validate the influence of social media on the participant’s attitude towards appearance (Appendix A). Items with the reply “Can’t say” were excluded, which explains the variable number of participants in the SATAQ sub-categories (pressure, awareness, internalization).

One-way ANOVAs found significant differences between groups in the three SATAQ-subcategories pressure (F(3,218) = 8.231, *p* < 0.001), awareness (F(3,218) = 7.901, *p* < 0.001), and internalization (F(3,218) = 6.968, *p* < 0.001). Thus, the control group reported significantly lower measures than most other groups (Figure 2).

Correlations between the SATAQ factors and F1 (α′ = 0.0125) found that in both online groups (SRPD, controls), percentual exacerbation of F1 significantly correlated with a heightened awareness, pressure, and overall SATAQ score. Additionally, in the control group, increase in F1 significantly correlated with internalization measures. Thus, those participants in the online samples who reported a higher susceptibility to content on social media also reported significantly higher increased disordered eating behaviours compared to those less susceptible to messages on social media.

### 3.5. Use and Helpfulness of Coping Strategies during COVID-19 Restrictions

The most helpful coping strategies when used were enjoyable activities, virtual contact with friends and family, relaxing and playing with the family, and day planning (Figure 3). View Appendix A for usage of each coping strategy and the overall number and helpfulness score by group. The helpfulness score consists of the sum of the 5-point Likert scale helpfulness scores of all 21 coping strategies divided by 21 and can range from 0 to 5. There was a significant difference in the number of coping strategies used (F(3,218) = 4.949, *p* < 0.01) and in the helpfulness score (F(3,218) = 9.348, *p* < 0.001) between the groups. Post hoc analysis revealed that patients with depression used significantly fewer coping strategies than the SRPD (*p* < 0.01) and control group (*p* < 0.01) and found the coping strategies overall significantly less helpful than the SRPD (*p* < 0.01) and control group (*p* < 0.001).

There was no significant correlation found between the number or helpfulness of coping strategies used and the percentual changes of the CIES factors.

### 3.6. Qualitative Analysis: Personal Experiences and Coping Strategies

Forty-three participants (19.4%) responded to the qualitative assessment of strategies and personal experiences. Participation was between 20 and 21% in all groups except for the depression group (5%). We identified “negative thoughts and mood”, “problems at home with family”, “struggles with body image and weight”, and “lack of social contacts” but also “positive experiences” as broad themes for personal experiences shared. For strategies, the categories found were “time with family and friends”, “working out”, “time spent outdoors and/or with animals”, “hobbies and interests”, and “positive thinking and relaxation exercises”. Exemplary quotes with group allocations are given in Table 3. The AN patients (*n* = 4) each mentioned walks or time spent in nature.

## 4. Discussion

In line with Ravens-Sieberer et al. [3], this present FRANCO study found that almost half of the adolescents reported a reduction of QoL during lockdown. In people with ED, QoL can be significantly increased by psychotherapy [33], which consequently highlights the need for availability of treatment even in times of social confinement.

In contrast to previous findings [19] but in line with Schlegl et al. [10], in AN patients (and the other groups), disordered eating deteriorated significantly during lockdown. Corresponding with hypothesis H1, analysis of changes in disordered eating, anxiety, and depressive symptoms and emotion regulation highlighted female adolescents with AN as a particularly vulnerable group that has been affected strongly during the pandemic. Since intolerance of uncertainty is highly prevalent in people with ED, especially in people with AN [34], the lockdown measures might have acted as a trigger mechanism for EDs. It has been shown that intolerance of uncertainty contributes to anxiety and depressive disorders [35] and positively correlates with depression, drive for thinness, and body dissatisfaction [36]. Thus, future research should explore the role of uncertainty in regard to its contribution to ED symptoms during significant changes in the lives of children and adolescents. In larger cohorts, vulnerability during certain age periods (early vs. later adolescence) and gender differences could be examined more specifically.

Anxiety, depression, and problematic emotion regulation significantly increased in all groups during lockdown. Consequently, the effects of social restrictions on the mental health of adolescents should remain closely monitored. H1 was only partly fulfilled since anxiety and depressive symptoms and problematic emotion regulation only significantly increased more in the AN group but not in the depression group compared to the other groups. As suggested by Schlegl et al. [10], considering the exacerbation of anxiety and depressive symptoms in adolescents with AN, interventions addressing these symptoms during lockdown might help manage ED symptomology.

In partial fulfilment of hypothesis H2, our findings suggest that female adolescents with AN (and SRPD) are particularly susceptible to the sociocultural body image portrayed on social media. Contrary to our hypothesis H2, in adolescents with AN (and depression), no significant correlation was found between susceptibility to messages on social media and increased disordered eating during the COVID-19 restrictions. Instead, in the online samples (SRPD, controls), higher receptibility to the sociocultural body ideal correlated with increased disordered eating symptomology. Thus, we suggest implementing large-scale social media literacy interventions for adolescents, e.g., in the school context, which have been shown to help reduce risk factors for EDs in adolescents, such as a negative body image and disordered eating [37], and may help manage struggles around social comparison during the pandemic. In the control group, younger age correlated with a higher increase of disordered eating during the pandemic, which indicates that interventions targeting disordered eating should be made available particularly for younger adolescents.

The coping strategies considered most helpful were time spent with family and friends in a phase of limited social contact. It was found that loneliness increases the risk for anxiety and depression in adolescents and young adults with pre-existing mental health problems [38]; thus, social isolation during lockdown might pose a risk for mental health problems. We suggest implementing interventions centred around conversation and social connection, as proposed by Fernández-Aranda et al. [29]. In contrast to hypothesis H3, whether or how successfully coping strategies were used did not correlate with changes of symptoms during lockdown. This could be because our survey may not have covered effective strategies to reduce the exacerbation of symptoms during lockdown. Adolescents might profit from positive thinking. Thus, a study among female undergraduate students found that negative problem orientation leads to worse coping during lockdown, more maladaptive coping behaviours, and less positive body image, suggesting viewing the pandemic as an opportunity for growth in order to prevent negative coping strategies [39]. Surprisingly, “interrupting thoughts and behaviours” rated least helpful among the strategies suggested. Analysis of use and helpfulness of coping strategies highlighted adolescents with depression as a group particularly in need for better-suited coping strategies. We suggest that future studies should investigate adequate coping strategies that would effectively help adolescents manage changes to their daily routines and social life.

Results of qualitative data analysis pointed out problems at home with the family during lockdown, which a meta-analysis of 21 papers found can upkeep ED symptomology, especially in adolescents [40]. Future research should focus on family factors, e.g., by including questionnaires on the family system, family communication style, and parents′ reports. Family interventions might avert additional harm in the lives of adolescents. As for coping strategies, AN patients specifically reported walks in nature, which might function as a strategy for controlling weight rather than a coping strategy and may be partly related to poor insight, which has been found in previous studies in patients with AN [41].

AN patients have been affected particularly strongly during lockdown. We suggest family-based treatment, which is an evidence-based treatment for adolescents with AN [42], particularly for younger patients [43]. In times of social confinement, we propose virtual family-based treatment, which has been shown to significantly restore weight as well as improve depression and self-esteem measures in adolescents with AN [44]. Video-based psychotherapy has similar clinical outcomes to in-person therapy [45,46,47,48]. Furthermore, it has been suggested that this pandemic presents an opportunity to speed up the process of implementing telehealth [49,50]. However, it has been found that especially AN patients struggle to adapt to remote treatment [19]. Hence, programs tailored for the needs of AN patients in remote treatment settings might be necessary.

As suggested by Bruining et al. [51], the pandemic might also have a positive side effect, as some adolescents reported positive personal experiences. Yet, taken together our FRANCO study, results suggest that the COVID-19 lockdown measures were perceived as a negative experience in the lives of adolescents, especially when afflicted with AN.

### Limitations

There are serious limitations to consider, such as a limited sample size, group differences in age, and lack of male and non-binary participants. Future studies should explore how male AN patients, boys in general, and non-binary adolescents for whom concerns about eating and body shape are also a growingly important issue might be affected by COVID-19 restriction measures. Online recruitment limits accessibility of the survey to only adolescents with an online presence. For participants to compare how they felt before and during lockdown required them to think back, and answers were therefore susceptible to recall bias. For future research, longitudinal studies should be considered. Socially desirable responding might lead to false self-reported information. The contribution of lockdown to the change of symptomology cannot be differentiated from other factors, such as growing older, (lack of) treatment, and other factors influencing the course of disease in clinical samples. A lack of usage of a qualitative data analysis program due to the quantitative focus of the research represents an additional limitation.

## 5. Conclusions

The present study provides information on how female adolescents experienced the COVID-19 lockdown measures. The results point out the importance to particularly consider adolescents with AN since they reported to have been strongly affected by the COVID-19 lockdown measures regarding increase of disordered eating and anxiety and depressive symptoms. In non-clinical samples (SRPD, controls), higher susceptibility to body image content on social media and a reported increase of disordered eating during COVID-19 lockdown correlated. Almost half of all participants reported a reduced QoL due to COVID-19 restriction measures. Future studies should explore long-term effects of confinement measures on the mental health of adolescents and better-fitted coping strategies.

## Figures and Tables

**Figure 1 ijerph-19-08480-f001:**
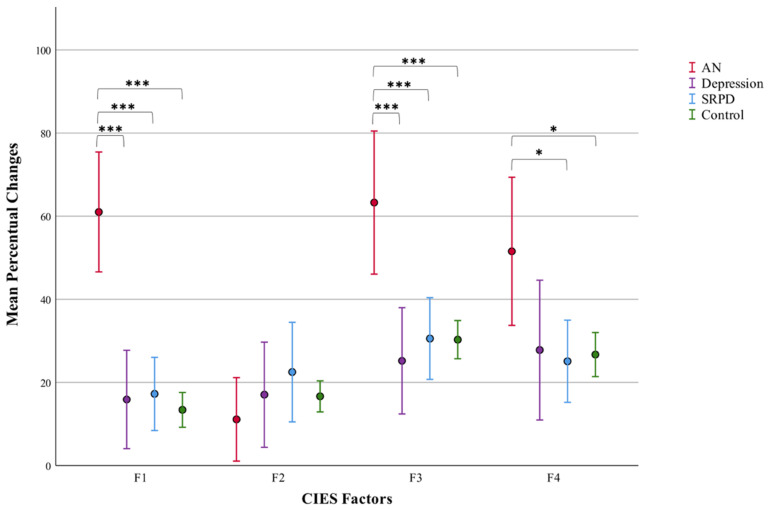
Percentual changes of CIES factors pre vs. during lockdown by group with 95% confidence interval error bars; results of ANOVAs with pairwise post hoc Bonferroni corrections for number of tests (* *p* < 0.05, *** *p* < 0.001) (F1, restrictive eating; F2, overeating; F3, anxiety and depressive symptoms; F4, emotion-regulation problems).

**Figure 2 ijerph-19-08480-f002:**
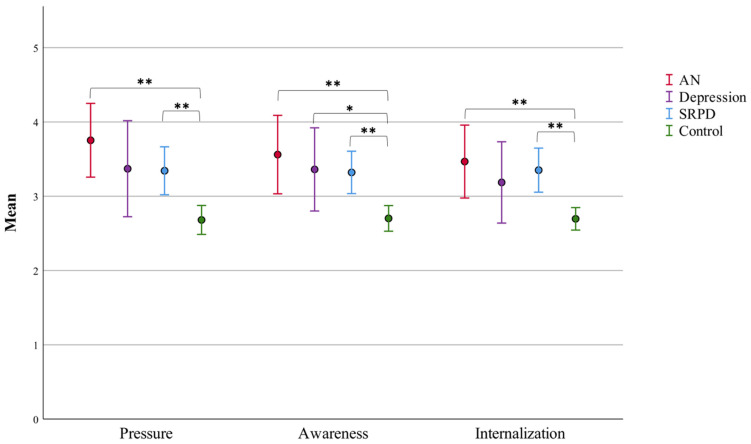
SATAQ scores by group with 95% confidence interval error bars; results of ANOVAs with pairwise post hoc Bonferroni corrections for number of tests (* *p* < 0.05, ** *p* < 0.01) (SATAQ, Sociocultural Attitudes Towards Appearance Questionnaire).

**Figure 3 ijerph-19-08480-f003:**
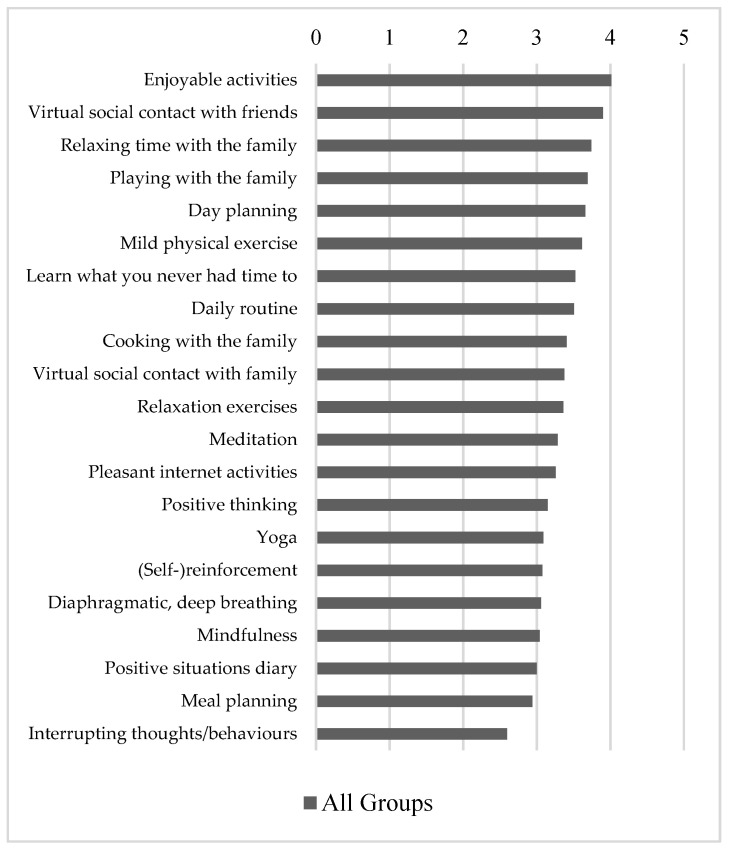
Responses on helpfulness of coping strategies (when used) for all groups (1 = not helpful, 2 = a little helpful, 3 = fairly helpful, 4 = helpful, 5 = very helpful) sorted by degree of helpfulness.

**Table 1 ijerph-19-08480-t001:** Means and standard deviations (SDs) of age and BMI (body mass index) by group (AN, anorexia nervosa; SRPD, self-reported psychiatric diagnosis; BMI, body mass index).

	All Groups (N = 222)	AN (N = 19)	Depression (N = 20)	SRPD (N = 45)	Control (N = 138)
**Age (years)**					
Mean (SD)	15.53 (1.92)	14.60 (1.31)	15.60 (1.28)	16.28 (1.89)	15.40 (2.00)
Range	11.2–18.9	12.5–17.6	13.5–17.9	12.3–18.7	11.2–18.9
**BMI (kg/m^2^)**					
Mean (SD)	19.89 (3.51)	16.91 (1.88)	21.05 (3.23)	21.02 (4.69)	19.76 (3.01)
Range	12.74–39.18	12.74–19.82	16.51–30.76	14.5–39.18	14.06–31.57

**Table 2 ijerph-19-08480-t002:** Percentual distribution of responses on worsening of perceived quality of life due to COVID-19 restrictions by group.

	Strongly Agree %	Agree %	Undecided %	Disagree %	Strongly Disagree %	Cannot Say %
All groups	14.0%	33.8%	11.7%	18.0%	15.2%	7.2%
AN	15.8%	52.6%	10.6%	5.3%	5.3%	10.5%
Depression	35.0%	35.0%	10.0%	5.0%	10.0%	5.0%
SRPD	20.0%	33.3%	20.0%	13.3%	8.9%	4.4%
Control	8.7%	31.2%	9.4%	23.2%	19.6%	8.0%

**Table 3 ijerph-19-08480-t003:** Qualitative assessment of personal experiences and strategies with group allocation.

**Personal Experiences**	
Negative thoughts and mood	“Ultimately, the entire lockdown was a development stage that felt like hell though, especially because one is constantly confronted with oneself.” (SRPD) “It’s very hard not to think negatively in this time [of lockdown].” (Control) “Unfortunately, I fell into a big hole again once the school had started again (…) I didn’t manage to get out of bed anymore or to motivate myself for anything.” (SRPD)
Problems at home with family	“(…) the isolation in the same house with the same people was just exhausting.” (Control) “a hard test for our family life” (Control)
Struggles with body image and weight (weight loss, weight gain)	“I ate a lot and also gained weight and am very discontent with that now. Because of social media I don’t know if I want to learn to accept myself or lose weight. I’m indecisive what would make me happier.” (Control) “I lost over 10 kg in one year. I was a little bigger before but definitely more carefree and more confident than I am now. I miss my old self. Since the gyms have opened up again, I go there until I must grant myself a rest day. I love exercising, it’s not compulsive in that sense, rather the constant thought about what I’m going to eat today.” (SRPD)
Lack of social contacts	“practically sat at home all day” (Control)
Positive experiences	“The lockdown was good for me, because going to school is very hard for me. (…) I started becoming more confident and worked on myself, which caused me to feel much better.” (SRPD) “All in all, I experienced many good things and took more time for things. I was more motivated, and I think that the lockdown was good for me.” (Control)
**Strategies**	
Time with family and friends	“It also helped me a lot to talk to my friends on the phone.” (Control) “I especially used the time to do more with my family, particularly my siblings.” (SRPD)
Working out	“When I was bored, I tried not to eat anything, but to work out.” (Control) “By exercising regularly, I added variety into my everyday life.” (SRPD)
Time spent outdoors in nature and/or with animals	“walks in nature” (AN) “walks with my dog” (AN) “Contact with animals helped me a lot.” (Control)
Hobbies and interests	“tried new hobbies” (SRPD) “watching movies and TV-shows, playing violent computer games” (Depression)
Positive thinking, relaxation exercises	“My number one coping strategy was self-reflection, trauma processing, inner child-healing” (SRPD) “breathing exercises” (Control) “One should always think positive even when the body changes” (SRPD) “Always try to think positive, but that doesn’t help much with body insecurities.” (Control)

## Data Availability

The data presented in this study are available on request from the corresponding author.

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
