# Peer review of "Effects of the COVID-19 Restrictions on Eating Behaviour and Eating Disorder Symptomology in Female Adolescents"

_ijerph, 2022, doi:10.3390/ijerph19148480_

Round 1

Reviewer 1 Report

I thank the editor for the opportunity to review this paper. I congratulate the authors for choosing the topic, which is very interesting and current.

Introduction: Introduction: I consider it very good and sufficient regarding the topic addressed.

Methods: It is necessary to better describe how the selection of participants was made, as it was necessary that some participants had psychiatric diagnoses, but were recruited online through social media, email, etc. Better describe the criteria for choosing the control group, including the number of participants in this group. I suggest entering more details about the instruments applied, such as: for which population the instruments were validated and analysis variables should be clearer. Enter more details about the qualitative analysis of the data.

Results: The test applied should be included in the footer of the tables, as well as improving the title of each one of the tables, specifying the study population and what was compared. The description of the results of the figures can be clearer.

Limitations: Include online recruitment and use of online questionnaires as it may limit survey access to social media users. Include the lack of standardization and use of a qualitative data analysis program.

Conclusion: Summarize the main findings

Reviewer 2 Report

Thanks to the editor for the invitation. In this study, the authors have investigated the association between COVID-19 Restrictions and Eating Behaviour and Eating Disorder Symptomology in Female Adolescents. This is an interesting study, I appreciate the author's idea whereas there may be some problems with the design of this study.

 Please see my comments below.

1.       Is this a case-control study? If yes, it is better to match the cases and control group by potential risk factors (e.g., age or the duration of COVID-19 Restrictions)

2.       Is there some information on duration of COVID-19 Restrictions?

3.       Are the participants infected with COVID-19? The health status may also affect the eating disorder.

4.       It is unusual to see six hypotheses in one study. You may use one or two hypotheses to summary them.

5.       Is there any approach to verify the accuracy of self-reported information?

Reviewer 3 Report

Thank you for the opportunity to review this paper exploring the effects of COVID-19 restrictions on eating disorder symptomatology in female adolescents.

The paper is generally well-written, and methodology is clearly described.

Minor issues for improvement prior to publications are as follows:

1. Inclusion of further description regarding the duration/timing of lockdowns in Germany and how this relates to when data was collected (so that readers in other countries can know how applicable results are to their populations).

2. Explanation of why boys were left out, and addition of this as a study limitation.

3. Although qualitative data is presented, the authors state that no formal coding scheme was used.  However, clearer description of who coded and how presented themes were identified is needed. It sounds like latent pattern content analysis was used, but this may also not be the case.  

4. The potential influence of family/systemic factors to young people's symptoms and well-being is underplayed, and would be worth discussing further - especially as there is some mention of it in the qualitative feedback.

5. Some useful recommendations are made in the discussion, but probably extend beyond the actual study findings.  For example, recommending ACT in response to the H1-related finding that young people with eating disorders were more vulnerable during the pandemic.  It would be better to state recommendations for follow-up research (e.g. exploring potential mediators, such as uncertainty).  

6. Despite no association being found between susceptibility to social media messaging and eating disorders, the authors recommend implementation of large-scale social media literacy interventions.  On what basis?  Once again, it seems that recommendations exceed study findings.

 I hope that these issues can be rectified prior to publication.

Round 2

Reviewer 1 Report

I appreciate the changes made and consider that the paper is ready to be published.  Congratulations to the authors.

Author Response

Thank you.

Reviewer 2 Report

Great response! I don't have other comments.

Author Response

Thank you.